# Genomic and Functional Regulation of TRIB1 Contributes to Prostate Cancer Pathogenesis

**DOI:** 10.3390/cancers12092593

**Published:** 2020-09-11

**Authors:** Parastoo Shahrouzi, Ianire Astobiza, Ana R. Cortazar, Verónica Torrano, Alice Macchia, Juana M. Flores, Chiara Niespolo, Isabel Mendizabal, Ruben Caloto, Amaia Ercilla, Laura Camacho, Leire Arreal, Maider Bizkarguenaga, Maria L. Martinez-Chantar, Xose R. Bustelo, Edurne Berra, Endre Kiss-Toth, Guillermo Velasco, Amaia Zabala-Letona, Arkaitz Carracedo

**Affiliations:** 1Center for Cooperative Research in Biosciences (CIC bioGUNE), Basque Research and Technology Alliance (BRTA), Bizkaia Technology Park, Building 801A, 48160 Derio, Spain; pshahrouzi@cicbiogune.es (P.S.); iastobiza@cicbiogune.es (I.A.); acortazar@cicbiogune.es (A.R.C.); vtorrano@cicbiogune.es (V.T.); amacchia@cicbiogune.es (A.M.); imendizabal@cicbiogune.es (I.M.); aercilla.ciberonc@cicbiogune.es (A.E.); lcamacho@cicbiogune.es (L.C.); leirearreal@gmail.com (L.A.); mbizcarguenaga@cicbiogune.es (M.B.); mlmartinez@cicbiogune.es (M.L.M.-C.); eberra@cicbiogune.es (E.B.); azabala@cicbiogune.es (A.Z.-L.); 2CIBERONC (Centro de Investigación Biomédica en Red de Cáncer), 28029 Madrid, Spain; ruben.fdez.caloto@gmail.com (R.C.); xbustelo@usal.es (X.R.B.); 3Biochemistry and Molecular Biology Department, University of the Basque Country (UPV/EHU), P.O. Box 644, E-48080 Bilbao, Spain; 4Medicine and Surgery Department, Veterinary Faculty, Complutense University of Madrid, 28040 Madrid, Spain; jflores@vet.ucm.es; 5Department of Infection, Immunity and Cardiovascular Disease, University of Sheffield, Beech Hill Road, Sheffield S10 2RX, UK; c.niespolo@sheffield.ac.uk (C.N.); e.kiss-toth@sheffield.ac.uk (E.K.-T.); 6Centro de Investigación del Cáncer, Instituto de Biología Molecular y Celular del Cáncer, Consejo Superior de Investigaciones Científicas (CSIC)-University of Salamanca, 37007 Salamanca, Spain; 7Centro de Investigación Biomédica en Red de Enfermedades Hepáticas y Digestivas, CIBERehd, Carlos III Health Institute, 28029 Madrid, Spain; 8Biochemistry and Molecular Biology Department, Complutense University of Madrid, 28040 Madrid, Spain; gvelasco@quim.ucm.es; 9Instituto de Investigaciones Sanitarias San Carlos, 28040 Madrid, Spain; 10Ikerbasque, Basque Foundation for Science, 48011 Bilbao, Spain

**Keywords:** TRIB1, prostate cancer, cMYC, mouse models

## Abstract

Prostate cancer is the most frequent malignancy in European men and the second worldwide. One of the major oncogenic events in this disease includes amplification of the transcription factor cMYC. Amplification of this oncogene in chromosome 8q24 occurs concomitantly with the copy number increase in a subset of neighboring genes and regulatory elements, but their contribution to disease pathogenesis is poorly understood. Here we show that *TRIB1* is among the most robustly upregulated coding genes within the 8q24 amplicon in prostate cancer. Moreover, we demonstrate that *TRIB1* amplification and overexpression are frequent in this tumor type. Importantly, we find that, parallel to its amplification, *TRIB1* transcription is controlled by cMYC. Mouse modeling and functional analysis revealed that aberrant TRIB1 expression is causal to prostate cancer pathogenesis. In sum, we provide unprecedented evidence for the regulation and function of TRIB1 in prostate cancer.

## 1. Introduction

The pathogenesis of cancer is underscored by mutations in driver genes that support the acquisition of cancer hallmarks [1,2]. Copy number aberrations can affect a single gene, a group of coding and non-coding genes or DNA regulatory regions [3]. The genomic locus containing the oncogene *cMYC*, 8q24, is an illustrative example of broad regulatory impact of genomic aberrations [4,5]. *cMYC* is frequently amplified in tumors [4,6]. Whereas focal amplifications in this gene are reported, copy number alterations in this locus often encompass neighboring regulatory regions, coding and non-coding genes [5,7]. Indeed, a number of genes contained in the cMYC locus have been involved in the pathogenesis or progression of different cancers, including *BOP1*, *PVT1*, *FAM84B* or *POU5F1P1* [7,8,9,10,11,12,13]. However, a comprehensive analysis of *cMYC*-neighboring genes in specific tumor types is lacking, thus resulting in an incomplete understanding of the molecular drivers of this disease. 

Prostate cancer (PCa) is among the most frequent cancer types in men, and it is responsible for an important fraction of cancer-associated mortality [14]. This disease is predominantly diagnosed in a localized stage and is subject to first-line therapies, including prostatectomy and radiotherapy [15,16]. However, a subset of patients will exhibit a raise in blood prostate-specific antigen (PSA) months to years after treatment, which is indicative of disease recurrence. Albeit the implementation of innovative therapies for recurrent PCa, emergence of metastasis in these patients is frequent, which represents a major risk of mortality by this disease. 

cMYC is a well-known driver of PCa pathogenesis and progression [4]. This gene is frequently amplified and upregulated in PCa, and an increase in cMYC dosage has been reported to associate with disease progression and castration-resistant PCa [17]. Increased expression of this transcription factor is an initiating event in this disease, as demonstrated in genetic mouse models [18,19]. Importantly, cMYC overexpression cooperates with other genetic perturbations, to promote disease progression [18,19]. Interestingly, despite its frequent upregulation, a recent report ruled out a significant prognostic value of cMYC protein levels when monitoring lethality as the outcome [20]. 

The Tribbles (TRIB) proteins are a family of serine/threonine pseudokinases composed of three members, TRIB1, TRIB2 and TRIB3 [21,22]. TRIB family proteins are activated by a number of cellular stresses and mitogens, and have been reported to participate in cancer-related processes [22]. Their lack of catalytic activity has inspired various studies aimed at identifying their molecular mechanism of action. These pseudokinases harbor a C-terminal COPI-binding domain, which controls the stability of interacting proteins, through ubiquitination and proteasome-dependent degradation [22,23,24,25,26]. The three members of the family operate as tumor suppressors or tumor promoters, based on the tissue of origin [21,22]. 

*TRIB1* gene localizes to chromosome 8q24.13, in close proximity to *cMYC*. Amplifications of this gene are reported in cancer [22], and evidence of its contribution to disease pathogenesis has begun to emerge. For example, TRIB1-mediated degradation of C/EBPα through COPI and activation of MAPK/AKT pathways lead to leukemogenesis [22]. With regards to PCa, the evidence on the function of TRIB1 is limited, and no genetically engineered mouse models have been generated to provide formal demonstration of its tumor-promoting activity [27,28,29]. 

In this study, we demonstrate that *TRIB1* is the gene exhibiting the highest expression within *cMYC* amplicon in PCa. In addition to the co-amplification, the pseudokinase is also a transcriptional target of cMYC. Importantly, we show that exacerbated expression levels of *Trib1* contribute to the pathogenesis of PCa in murine models.

## 2. Methods

### 2.1. Animals

All mouse experiments were performed by following the ethical guidelines established by the Biosafety and Animal Welfare Committee at CIC bioGUNE, Derio, Spain (under protocol P-CBG-CBBA-0715). The employed procedures followed the recommendations from the Association for Assessment and Accreditation of Laboratory Animal Care International (AAALAC). Genetically engineered mouse model experiments were performed in a mixed background, as reported [30]. The prostate-specific *Pten*-deficient mice were originally generated by the Pandolfi group [31,32]. Mice were routinely fasted for 6 h prior to tissue harvest (9:00–15:00), to prevent metabolic alterations due to immediate food intake. To address the effect of *Trib1* overexpression on PCa pathogenesis, *Rosa26LSL-Trib1^Tg^* mice [33] were crossed with *Pten^lox/+^ Pb-Cre4* mice. 

For xenograft assays, 4 × 10^6^ DU145 cells transduced either with empty TRIPZ vector (mock) or TRIB1 expressing construct (Doxycycline-inducible TRIPZ–TRIB1) were prepared in PBS supplemented with 5 mM glucose. Matrigel (Corning Cat# 354230) was mixed with the cell suspension, at a 1:1 ratio, in a final volume of 100 µL, and injected subcutaneously in two flanks per mouse (8 mice, *n* = 16 per condition) in immunocompromised male nude mice of 8–10 weeks (Harlan). Mice were randomly assigned to doxycycline or control diet [30] at day 4 after injection. Tumor size was monitored every day, using external caliper, during a total of 29 days. Tumor volume was inferred by using the volume estimation of an ellipsoid. At the experimental endpoint, mice were sacrificed, and tumors were processed for molecular analysis. 

### 2.2. Histopathological Analysis and Immunohistochemistry

Tissue sample collection was carried out at 15–17 months of age (*Pten^pc+/−^ Trib1^pc+/+^ and Pten^pc+/−^ Trib1^pcTg/+^* mice). Tissue samples were fixed overnight in 10% neutral buffered formalin, embedded in paraffin and sectioned 3 μm thick and dried. Slides were dewaxed and re-hydrated through a series of graded ethanol until water and subsequently stained with required antibody and/or hematoxylin–eosin (H&E). Histological observations on H&E stained tissues were performed, using an Olympus DP73 digital camera. Prostate lesions were histologically classified according to the criteria of the Consortium Prostate Pathology Committee [34] and scored as follows: 0 = no lesion observed; 1 = focal or multifocal LGPIN (low-grade prostatic intraepithelial neoplasia); 2 = focal or multifocal HGPIN (high-grade prostatic intraepithelial neoplasia); 3 = focal carcinoma (less than 50% of tissue); 4 = invasive carcinoma (more than 50% of tissue). Ki67 (Ventana, ref. 790-4286, ready-to-use nuclear staining) and F4/80 (BioRad-MCA497) staining were performed in automated immunostainers (BenchMark Ultra, Ventana Medical Systems, Tucson, AZ, USA), following routine methods. Tris-EDTA was used for antigen retrieval. The analysis was performed by using a Nikon Eclipse 80i microscope (Tokyo, Japan). 

### 2.3. Cell Culture

Human prostate carcinoma cell lines PC3, DU145 and LnCaP were purchased from Leibniz-Institut DSMZ-Deutsche Sammlung von Mikroorganismen und Zellkulturen GmbH, who provided the authentication certificate. Human prostate cell lines PWR1E, RWPE1 and BPH1 and human prostate carcinoma cell lines 22RV1 and VCaP were purchased from American Type Culture Collection (ATCC). HEK293FT were purchased from Thermo Fisher and used for lentiviral production and lipofectamine-based transient transfection. C4-2 was generously provided by the laboratory of Dr. Pier Paolo Pandolfi. Cell lines were periodically subjected to microsatellite-based identity validation. None of the cell lines used in this study was found in the database of commonly misidentified cell lines maintained by the International Cell Line Authentication Committee and NCBI Biosample. All cell lines were routinely monitored for mycoplasma contamination. DU145, PC3, VCaP and HEK293FT cell lines were maintained in DMEM (Gibco Cat# 41966-029) media supplemented with 10% Fetal Bovine Serum (FBS; Gibco) and 1% penicillin–streptomycin (Gibco; 10,000 U/mL). LNCaP, C4-2 and 22RV1 cell lines were maintained in RPMI media (Gibco Cat# 61870-010; with GlutaMAX supplement) supplemented with 10% FBS and 1% penicillin–streptomycin. PWR1E, RWPE1 and BPH1 cell lines were maintained in Keratinocyte Serum Free Medium (K-SFM; Gibco) supplemented with 0.05 mg/mL Bovine Pituitary Extract (BPE; Gibco) and 5 ng/mL epidermal growth factor (EGF; Gibco). 

### 2.4. Generation of Stable Cell Lines

TRIB1-HA and TRIB1 were cloned into TRIPZ^TM^ vector as previously reported [30].

Lentiviral vector expressing a validated shRNA against human *cMYC* (TRCN0000039642) or *TRIB1* (TRCN0000381401) from the Mission shRNA Library was subcloned in a Tet-pLKO inducible system (Addgene plasmid # 21915) kindly donated by Dr. Wiederschain [35]. Cells were transfected with lentiviral vectors, following standard procedures [30,36], and viral supernatant was used to infect cells. Selection was done by using puromycin (2 µg/mL) or blasticidin (10 µg/mL), as required.

### 2.5. Cellular Assays

Two-dimensional cell growth, anchorage-independent growth and invasive growth were performed as previously reported [30,36]. For colony-formation assay, 500 cells/well were seeded in a 6-well plate. The cells were allowed to grow and form foci for up to 14 days. After this period, cells were washed and fixed with formalin, and further stained with crystal violet [30]. The plates were scanned for counting the number of colonies with Image J. Then, 1 mL of acetic acid was added to each well and allowed the crystal violet to dissolve. Afterward, 75 µL of the solution was transferred to 96-well plates, and absorbance was measured at 590 nm.

### 2.6. Real-Time Quantitative PCR 

RNA was extracted using NucleoSpin^®^ RNA isolation kit (Macherey-Nagel; ref: 740955.240C). For murine tissues a Trizol-based implementation of the NucleoSpin^®^ RNA isolation kit protocol was used, as referenced [37]. For all cases, 1 μg of total RNA was used for cDNA synthesis, using Maxima^TM^ H Minus cDNA Synthesis Master Mix (ThermoFisher, M1682). Quantitative Real-Time PCR (RT-qPCR) was performed as previously described [30,38]. Universal Probe Library (Roche) primers and probes employed (Roche; Thermo Fisher) are detailed in Appendix A. All RT-qPCR data presented were normalized by using *GAPDH/Gapdh* (Applied Biosystems; Hs02758991_g1, Mm99999915_g1) and/or *ß-ACTIN/ß-Actin* (Hs99999903-m1, Mm00607939_s1). The majority of assays was performed by using 2 independent housekeeping genes with consistent results, but data with one normalizer are shown for simplicity.

### 2.7. Western Blot

Western blot was performed as previously described [36]. Briefly, cells were lysed in RIPA buffer (50mM TrisHCl pH 7.5, 150 mM NaCl, 1mM EDTA, 0.1% SDS, 1% Nonidet P40, 1% sodium deoxycholate, 1 mM Sodium Fluoride, 1 mM sodium orthovanadate, 1 mM β-glycerophosphate and protease inhibitor cocktail; Roche). Antibodies used are described in Appendix A. Mouse and rabbit secondary antibodies were purchased from Jackson ImmunoResearch. After standard SDS-PAGE and Western blotting techniques, proteins were visualized, using the ECL (enhanced chemiluminescent) in iBright (Thermo Fisher).

### 2.8. Chromatin Immunoprecipitation

Chromatin immunoprecipitation (ChIP) was performed as previously reported [36], using the SimpleChIP Enzymatic Chromatin IP Kit (catalog no. 9003, Cell Signaling Technology, Inc). Briefly, 4 million PC3 cells per immunoprecipitation were grown in 150 mm dishes. Cells were cross-linked with 37% formaldehyde, for 10 min, at room temperature. Glycine was added to dishes, and cells were incubated for 5 min, at room temperature. Cells were then washed twice with ice-cold PBS and scraped into PBS and 200X Protease Inhibitor Cocktail (PIC). Pelleted cells were lysed, and nuclei were harvested, following the manufacturer’s instructions. Nuclear lysates were digested with micrococcal nuclease for 20 min, at 37 °C, and then sonicated in 500 mL aliquots, on ice, for six pulses of 20 s, using a Branson sonicator. Cells were held on ice for at least 20 s between sonications. Lysates were clarified at 11.000 g for 10 min, at 4 °C, and chromatin was stored at 80 °C. Anti-c-MYC antibody (Cell Signaling Technology #5605) and IgG antibody (Cell Signaling Technology #2729) were incubated overnight (4 °C) with rotation, and protein G magnetic beads were incubated for 2 h (4 °C). Washes and elution of chromatin were performed while following manufacturer’s instructions. DNA quantification was carried out, using a Viia7 Real-Time PCR System (Applied Biosystems) with SYBR Green reagents and primers that amplify a c-MYC binding region on *TRIB1* promoter (Primer information in Appendix A). 

### 2.9. Dual Luciferase Reporter Assay

*TRIB1* promoter region containing two cMYC binding sites (chr8:126441287-126441960 and chr.8:126442208-126442754) was cloned into pGL3-Firefly vector. pWZL-cMYC was a gift from William Hahn (Addgene plasmid # 10674) [39] and was used for overexpression of cMYC. Then, 15.000 HEK293FT cells were transiently transfected, using Lipofectamin^®^ 2000 (ThermoFisher) according to manufacturers’ indications with pGL3-TRIB1 promoter-Firefly (0.07 μg), empty or pWZL-cMYC (0.02 μg) and Renilla-expressing vector (5 μg) in a 96-well plate. After 24 h, the luciferase activity of both Firefly and Renilla was measured by a luminometer, using a dual luciferase assay reagent (Promega), and the ratio of Firefly to Renilla was calculated. Total cellular extracts were analyzed by Western blot, to confirm cMYC overexpression. 

### 2.10. Bioinformatics Analysis

The analysis of integration of copy number aberrations and gene expression in PCa TCGA (PRAD) was performed as follows. TCGA-PRAD cohort RNAseq counts were downloaded from Genomic Data Commons (GDC) server, using TCGAbiolinks R package, and further processed: Outlier samples were removed, low-expressed genes were filtered out and data were normalized (EDASeq-powered function). Finally, a differential expression analysis (DEA) was performed between tumor and normal samples, and *cMYC* amplicon differentially expressed genes (DEGs) were retrieved (|Log2(FC)| > 0.58 and FDR-value <0.05). GISTIC2.0 thresholded-by-gene data were downloaded from Broad’s Institute Firehose database latest run, using RTCGAToolbox R package. Then, we calculated differentially expressed genes between copy-number-altered (deep amplified/deleted) vs. diploid tumor samples for every gene contained in *cMYC* amplicon (|Log2(FC)| > 0.58 and FDR-value < 0.05).

The patient gene expression dataset analysis was performed, using CANCERTOOL [40]. In the microarray data, where gene expression was represented by various individual probes, the average of their signals was calculated and represented. Pearson correlation test was applied to analyze the correlation between paired genes. The *p*-value in these analyses indicates the significance of Pearson’s *r* coefficient. For the DFS analysis, patients were separated into the four different quartiles regarding its gene expression levels. In the case of signatures, the average of their gene expression levels was calculated. Kaplan–Meier Estimator [41] was used to estimate the survival curves of the different groups, while a Log-Rank test [42] was used to provide the *p*-value. Patient copy number information was obtained from cBioPortal [43,44] and TCGA Copy Number Portal (http://portals.broadinstitute.org/tcga/home). GISTIC analysis was performed on TCGA copy number data from version 3.0 of the SNP pipeline, on 20-Feb-2014, where 28 cancer types and 8663 tumor samples were analyzed by employing the stddata__2014_02_15 TCGA/GDAC tumor sample sets from FireHose (Appendix A). Visualization of the genomic position of genes in the *cMYC* locus (chr8:119897767-129710968) and their copy number status in the TCGA downloaded from cBioPortal was performed using gviz package [45]. ENCODE 3 data were analyzed via the UCSC genome browser (https://genome.ucsc.edu). Specifically, we explored the table “wgEncodeRegTfbsClusteredV3” containing ChIP-seq clusters, representing combined signals for 130 cell types. DNA binding motifs were obtained from ENCODE Factorbook repository.

### 2.11. Statistics Analysis and Reproducibility 

No statistical method was used to predetermine sample size. The experiments were not randomized. The investigators were not blinded to allocation during experiments and outcome assessment. Unless otherwise stated, data analyzed by parametric tests are represented by the mean ± SEM of pooled experiments and median ± interquartile range for experiments analyzed by non-parametric tests. The n-values represent the number of independent experiments performed, the number of individual mice or patient specimens. For each independent in vitro experiment, at least three technical replicates were used, and a minimum number of three experiments were performed, to ensure adequate statistical power (the number of biological replicates is indicated in the figure legends). In the in vitro experiments, normal distribution was assumed, and one-sample *t*-test was applied for one-component comparisons with control and Student’s *t*-test for two-component comparisons. Student’s *t*-test was used to compare data with normal distribution, and non-parametric Mann–Whitney exact test was used for samples not following a normal distribution. The confidence level used for all the statistical analyzes was of 95% (alpha value = 0.05). Two-tailed statistical analysis was applied for experimental design without predicted results, and one-tail for validation or hypothesis-driven experiments. GraphPad Prism 8.0.2 software and R version 3.6.0 were used for statistical calculations. 

## 3. Results

### 3.1. Identification of PCa-Relevant Candidate Genes in cMYC Amplicon

We aimed at studying the genes contained in *cMYC* amplicon in PCa. To this end, we took advantage of the TCGA copy number portal, which allows the analysis of copy number alterations in 28 cancer types and 8663 tumor samples through a simplified interface (http://portals.broadinstitute.org/tcga/home). Using this resource, we established that *cMYC* amplicon encompasses a genomic region containing 60 genes in PCa (Figure 1A and Appendix A). We next studied whether amplification of these genes was associated with increased expression in prostate tumors. Two this end, we performed two complementary analyses. On one hand, we integrated genomic amplification and gene expression data from TCGA (PCa, PRAD, Figure 1B). From the genes contained in *cMYC* amplicon, only 15 exhibited a significant upregulation concomitant to the amplification, whereas *ANXA13* and *COL14A1* exhibited unexpected repression (Figure 1B). On the other hand, we ascertained the expression levels in localized PCa compared to normal prostate specimens in five different patient datasets [40,46,47,48,49,50]. We established two stringent criteria to identify PCa-relevant genes: (i) data available in at least three PCa datasets and (ii) consistent directional alteration in gene expression (significant in more than 50% of available PCa datasets). This analysis led to a shortlist of 10 genes (Figure 1C). Seven of these genes exhibited a consistent upregulation in PCa (including *cMYC*), whereas three exhibited a downregulation. From the seven genes overexpressed, four, apart from *cMYC*, were also shortlisted in the TCGA strategy (Figure 1B, *TRIB1, MAL2, PVT1* and *FAM84B*). This list contained genes previously associated with co-amplification with *cMYC* in cancer, such as *PVT1* or *FAM84B* [7,10,11,13,51], thus validating our strategy. Interestingly, we found that the pseudokinase *TRIB1* exhibited the highest overexpression among the selected genes, which encouraged us to study it further. The detailed gene expression analysis of *TRIB1* in the PCa datasets (localized PCa vs. normal tissue) is presented in Figure 1D and Appendix A. Of interest, the upregulation of *TRIB1* mRNA levels in prostate cancerous tissue is in line with the observations made by other groups at the protein level [27,28]. Next, we analyzed the frequency of amplification of *TRIB1* in PCa. A comprehensive analysis of copy number aberrations, using cBioPortal [43,44], confirmed the increased copy number of this gene in PCa (Figure 1E), similar to what is observed in *cMYC* (Appendix A). Of note, within this set of studies (2844 specimens, including primary tumor and metastases), 373 cases exhibited amplification in *cMYC* and/or *TRIB1*, and 85.5% of those presented co-occurrence in both genes (Fisher F, *p* < 0.001). Amplification of *TRIB1* was also detected in PCa cell lines, using the information contained in the Cancer Cell Line Encyclopedia, available in DepMap [52] (https://depmap.org/portal/, Appendix A), and its consequence on gene expression in PCa cell lines was analyzed by real-time quantitative PCR (RT-qPCR). Of note, 50% of the PCa cell lines evaluated exhibited a significant upregulation of the pseudokinase compared to benign cell lines (Figure 1F). We further corroborated that *TRIB1* amplification is frequent in other tumor types, as illustrated by the analysis of TCGA datasets (Appendix A). We extended this analysis to breast cancer, where we could detect a frequency of amplification in *cMYC* and *TRIB1* greater than 15% in two independent datasets (TCGA and METABRIC, Appendix A) [53,54]. Similar to the scenario in PCa, from this set of 2899 specimens profiled for copy number aberrations, 741 exhibited amplification in *cMYC* and/or *TRIB1*, and 88.1% exhibited co-occurrence (Fisher F, *p* < 0.0001). The overexpression of this gene in PCa was among the highest in all tumor types studied in TCGA, which reinforced the notion that this pseudokinase might be relevant for the biology of this tumor type (Appendix A). 

To ascertain the pathological context where *TRIB1* would be upregulated, we analyzed additional publicly available PCa datasets. Firstly, we evaluated the gene expression of *TRIB1* and *cMYC* in different pathological scenarios. To this end, we took advantage of a study that included benign prostate epithelial tissue from patients without PCa, together with PCa epithelial tissue and its adjacent normal and prostate intraepithelial neoplasia (PIN) lesions [55] (Appendix A). The results confirmed the upregulation of both *TRIB1* and *cMYC* in PCa epithelial tissue, compared to normal-adjacent epithelium and epithelium from normal specimens. Interestingly, the mRNA upregulation observed in PCa was recapitulated in PIN lesions (Appendix A). Secondly, we studied the alterations in *TRIB1* and *cMYC* in localized PCa vs. metastatic lesions. These two genes exhibited greater amplification in metastasis, compared to localized tumors (Appendix A) [56]. However, this event was not translated to elevated mRNA abundance in metastasis, compared to primary PCa (Appendix A) [46,47,48], suggesting that other levels of regulation at the epigenetic level might exist. Lastly, we studied whether the expression of *TRIB1*, *cMYC* or their combination could inform abut disease progression after prostatectomy. Neither mRNA expression of *cMYC* or *TRIB1* nor their combination exhibited prognostic potential in biochemical recurrence (Appendix A). 

### 3.2. cMYC Regulates the Expression of TRIB1 in PCa

When analyzing the association between *TRIB1* copy number and mRNA abundance in patient datasets, we interestingly observed that tumors with diploid *TRIB1* exhibited mRNA expression levels as high as biopsies categorized as *TRIB1* amplified cases (Figure 2A). This observation led us to hypothesize that additional mechanisms of *TRIB1* upregulation beyond the amplification could exist in PCa. We thus focused on the transcriptional regulation of this pseudokinase. To this end, we interrogated the promoter region of *TRIB1* in ENCODE3. We extracted PolR2A peaks located upstream (<1 kb) the transcriptional start site (TSS) of different TRIB1 transcripts. We subsequently ascertained the transcription factors that were associated with a high binding score (>600). Interestingly, we observed that cMYC was present in two different regions of the *TRIB1* promoter with the highest score (1000/1000), whereas two additional potential binding sites exhibited lower scores and were not considered for further analyses (Figure 2B and Appendix A). The two binding sites with high scores presented canonical and non-canonical E-boxes, suggestive of bona fide cMYC-regulated regions (Appendix A). To confirm these results in PCa, we performed chromatin immunoprecipitation (ChIP) in PC3 cells with anti-cMYC antibody, coupled to RT-qPCR-based quantification of the selected binding regions within the immunoprecipitate. As predicted, cMYC significantly bound to regions within the identified binding sites (Figure 2C, Appendix A). To validate the functional regulation of *TRIB1* expression by cMYC, we performed two complementary experiments. On the one hand, we carried out dual luciferase reporter assays using the promoter region of *TRIB1* that contained cMYC binding sites. Co-transfection of cMYC with Firefly-luciferase reporter system fused to *TRIB1* promoter in HEK293T cells resulted in significant increase in luciferase luminescence (Figure 2D, Appendix A). On the other hand, cMYC silencing with a previously validated doxycycline-inducible shRNA system [36,57] resulted in a significant decrease in *TRIB1* mRNA abundance in PC3 cells (Figure 2E). A similar effect was found in the breast cancer cell line MDAMB231 (a cell line which does not exhibit copy number alterations in *TRIB1* according to the Cell Line Encyclopedia [58]), thus suggesting that this might be a general mechanism of regulation (Appendix A). Altogether, our results demonstrate that cMYC is an unprecedented transcriptional regulator of *TRIB1* in PCa, thus providing a more comprehensive molecular perspective of the mechanisms underlying the overexpression of this pseudokinase in this disease.

### 3.3. TRIB1 does not Exhibit Cell-Autonomous Tumor-Promoting Activity in PCa Cell Lines

In order to ascertain the function of TRIB1 in PCa, we undertook an in vitro approach. Based on the gene expression analysis in cell lines (Figure 1F), we chose a low *TRIB1*-expressing cell line for the overexpression (Figure 3A,B) and a high-expressing cell line for the silencing (Appendix A) of the pseudokinase, respectively. Inducible TRIB1 expression (C-terminal HA-tagged or untagged) in DU145 cells did not alter consistently cell proliferation (Figure 3C). The expression of untagged TRIB1 significantly reduced cell number, whereas the C-terminal tag form of the pseudokinase did not exert any effect. Neither of the constructs altered colony formation (Figure 3D), anchorage-independent growth (Figure 3E) or invasive growth in three-dimensional systems (Figure 3F). Additionally, inducible *TRIB1* silencing in PC3 cells was inconsequential for the aforementioned parameters (Appendix A). These results argue against a cancer-cell-autonomous prominent function of TRIB1 in PCa. To acquire further insight about the tumor-promoting function of TRIB1 in cellular system, we took advantage of our low TRIB1-expressing DU145 cells in which we could activate the expression of ectopic TRIB1 through the use of doxycycline. We injected these cells in the flank of immunocompromised nude mice and activated the expression of the pseudokinase four days after implantation. In line with our in vitro results, ectopic TRIB1 expression did not elicit a significant effect on tumor growth (Appendix A). The results in this cell line are consistent with a recent report [27], and suggest that TRIB1 expression in PCa cells is inconsequential for tumor biology in the cell lines and conditions employed. It is worth noting that the lack of a fully functional stroma in immunocompromised mice, or the total lack of such a compartment in vitro could be important factors influencing the results.

### 3.4. Trib1 Overexpression Cooperates with Pten Heterozygosity to Promote PCa Pathogenesis

The biological insights on TRIB1 tumor-promoting activities are scarce. We sought to evaluate the impact of *Trib1* expression on prostate tumorigenesis by using well-characterized murine models. Deletion of the tumor suppressor *Pten* in the prostate epithelium results in PCa, whereas heterozygous loss is associated to the development of prostate intraepithelial neoplasia (PIN) [30,59,60]. We thus interrogated the expression of the pseudokinase in the prostate of mice with PCa (*Pten^pc−/−^*) vs. wild type counterparts (*Pten^pc+/+^*). Interestingly, mice with PCa exhibited elevated prostate *Trib1* gene expression (Figure 4A). The PTEN-PI3K pathway regulates the abundance of cMYC through diverse mechanisms [61,62,63,64,65]. Indeed, prostate-specific *Pten*-deficient mice exhibited elevated cMyc protein expression (Appendix A). As complementary evidence for the negative impact of *PTEN* on the mRNA expression of *TRIB1*, we interrogated the aforementioned human PCa datasets. In line with our observations in the murine model, *PTEN* expression negatively correlated with *TRIB1* mRNA abundance in various patient cohorts (Figure 4B). 

We have previously reported the generation of a Cre-dependent *Trib1* transgenic mouse model (*Rosa26-LSL-Trib1;* termed *Trib1^Tg^*) [33]. We took advantage of this experimental model in order to ascertain the contribution of the pseudokinase to PCa pathogenesis. To this end, we bred *Trib1^Tg/+^* females with *Pten^Lox/+^*, *Probasin (Pb)-Cre* males (Appendix A). We bred the resulting mice for at least three generations, to build a founder colony. From this cohort, we derived prostate-specific *Pten* heterozygous mice in which *Trib1* expression was elevated through the expression of the transgene (Figure 4C). Remarkably, aged prostate-specific *Pten* heterozygous mice (15–17 months old) expressing a transgenic copy of *Trib1* exhibited signs of invasion in the prostate tissue (Figure 4D,E and Appendix A). Detailed pathological analysis concluded that *Trib1* transgene increased PCa incidence in *Pten^pc+/−^* mice from 16.7% to 50%. These results provide a formal demonstration of the tumor-promoting activity of *Trib1*, using an unprecedented genetically engineered mouse model. The discrepancies between the genetic mouse model and the human cellular system (Figure 3 and Figure 5) could be due to the presence of a fully functional stroma in the former, or to intrinsic differences between human and murine PCa that could impact on the role and activity of TRIB1. 

Based on the pathological alterations associated to transgenic *Trib1* expression in the murine prostate, we evaluated biological alterations in vivo that would explain the phenotype. On the one hand, we measured epithelial cell proliferation in the two genotypes of interest, by means of Ki67 immunoreactive cell quantification (a clinically validated biomarker of cell proliferation [66]). As opposed to the in vitro phenotype, *Trib1* transgenic expression resulted in a significant increase in epithelial cell proliferation (Figure 5A,B). Of note, due to the different incidences of adenocarcinoma in the two genotypes, we cannot rule out that the proliferative phenotype emerges as a consequence of cancer initiation. Despite Ki67 being a clinically relevant marker of proliferation, it would be interesting to explore cell-cycle alteration, in further detail, with other markers. On the other hand, TRIB1 influences the polarization and infiltration of macrophages [27]. We studied the composition of the prostate stroma in the two genotypes of interest. In line with this notion, quantification of F4/80-positive cell in formalin-fixed and paraffin-embedded prostate tissue revealed a significant increase in macrophage-specific staining upon *Trib1* transgenic expression in our mouse model (Figure 5C,D). These results support the notion that *TRIB1* is deregulated through genomic and transcriptional alterations in PCa and promotes cancer pathogenesis in vivo.

## 4. Discussion

The reprogramming of transcriptional networks is a key event in cancer, in general, and in PCa, in particular [67]. *cMYC* is among the most prominent genes, altering the transcriptional makeup of tumor cells [4]. This oncogene is altered in cancer through multiple means, including genomic, transcriptional and post-transcriptional mechanisms [4]. In turn, cMYC regulates cell growth, survival, invasion and metabolism [4,68]. The central role of this transcription factor in cancer has led to research efforts focused on the identification of pharmacological means to inhibit its function [69]. Therefore, elucidating molecular pathways that are perturbed as a result of exacerbated cMYC activity represents a major focus in cancer research.

Biological alterations in cancer often stem from copy number aberrations that encompass a large set of genes. As for *cMYC*, its upregulation is frequently due to amplification of chromosome 8q24 [5] This locus is defined as a gene desert containing several non-coding RNAs and regulatory DNA regions [10]. Moreover, genomic alterations in this locus (translocations, insertions and single nucleotide polymorphisms) frequently fail to impact on *cMYC* expression, thus suggesting that this locus harbors other cancer-relevant elements [10]. We sought to elucidate coding genes that could be relevant to the biology of PCa by integrating genomics, transcriptomics and bioinformatics analysis. Our study revealed that, out of all the genes encoded in this region, only a small subset exhibited a consistent upregulation in PCa that would be in accordance with their amplification. Importantly, we validated the overexpression of previously reported genes in this locus, including *PVT* and *FAM84B* [10,11,13,70,71]. Our results also shed light on chromosome 8q24 genes that, despite their amplification, are profoundly repressed in PCa, such as *MTSS1*. This observation is in line with the documented epigenetic repression of this gene through different molecular means [72,73].

Interestingly, the analysis of chromosome 8q24 in PCa revealed that *TRIB1* is the most robustly upregulated gene, at a level comparable to *cMYC*. This gene belongs to a family of pseudokinases relevant for health and disease [22]. Aberrant expression of the three Tribbles family members has been associated to cancer pathogenesis and progression [22]. Genomic (amplification and microsatellite repeats) and epigenetic (microRNA-based regulation) alterations in the *TRIB1* gene are linked to cancer [22,28,29,74]. In this study, we showed that the association between *TRIB1* and *cMYC* spans beyond their co-localization to chromosome 8q24. We demonstrated that cMYC is an unprecedented transcriptional regulator of the pseudokinase, through at least two discreet genomic regions in *TRIB1* promoter that contain canonical and non-canonical E-Boxes. These results suggest that amplification of 8q24 locus has a double impact on *TRIB1* gene expression: (1) through the increase of its gene dosage and (2) through the upregulation of its upstream transcriptional activator *cMYC*. These data might explain the predominant overexpression of *TRIB1* among 8q24 genes in our PCa analysis. Interestingly, other oncogenic insults relevant to PCa also affect *TRIB1* gene expression. We found that deletion of *Pten* in the mouse prostate results in elevated mRNA abundance of the pseudokinase. Similarly, *PTEN* expression is inversely correlated with *TRIB1* in human PCa datasets. These results can be explained by the reported regulation of cMYC downstream the PI3K pathway [61,62,63,64,65] and reveal an interesting convergence of PCa-relevant oncogenic signals in the control of TRIB1. It remains to be investigated the repercussion of *TRIB1* transcriptional control by PTEN and cMYC in other pathophysiological conditions. 

An increasing body of evidence suggests that TRIB1 controls cellular functions associated to cancer aggressiveness [22,27,28,75,76]. On one hand, the regulation of proteasome-mediated control of protein stability by TRIB1 is a field of growing interest. TRIB1 protein contains a C-terminal COPI interacting domain that targets TRIB1-interacting proteins for ubiquitination and proteasome-dependent degradation [22,24,26]. Recent reports suggest that TRIB1 utilizes the substrate-recognizing region in the pseudokinase domain, to bring proteins in close proximity to COPI E3 ligase and hence promote their ubiquitination [23,24,25]. On the other hand, TRIB1 regulates the activation of oncogenic signaling pathways, such as MAPK and PI3K-AKT [22,77]. We performed a wide array of biological assays in PCa cell lines, upon genetic perturbation of *TRIB1* with tightly controlled molecular tools, and found that *TRIB1* overexpression or downregulation in vitro is largely inconsequential to two-dimensional and three-dimensional growth and invasion. These results are in contrast to reports by other groups in this tumor type [28,29], thus suggesting that further research is needed to define the molecular determinants of cell-autonomous TRIB1 activity in cancer cells. 

In vivo immune-competent mouse models are instrumental for the comprehensive study of cancer-relevant molecular events. These experimental systems recapitulate, to a greater extent, the biology of tumor cells in the context of a complete microenvironment, thus accounting for cell-autonomous and non-cell-autonomous regulation. Therefore, the development of genetically engineered mouse models is key to understand the molecular basis of cancer biology. Our group and others have contributed to the development and characterization of mouse models that are relevant for the study of PCa [30,31,60,78,79] or Trib1 function [33,80]. By taking advantage of a genetic setting of PCa susceptibility in mice (prostate-specific *Pten* heterozygosity) [59], we demonstrated that transgenic *Trib1* expression elicits a substantial increase in the incidence of PCa, thus providing unprecedented evidence in genetically engineered mouse models, to support the causal contribution of this pseudokinase to prostate tumorigenesis. Strikingly, and in contrast to our in vitro observations, *Trib1* overexpression in vivo significantly increased epithelial cell proliferation. These data, together with the alteration in the stroma composition in these mice (illustrated by the significant increase in tumor-infiltrating macrophages), suggest that a fraction of the tumor-promoting activity of TRIB1 might be associated to non-cell-autonomous effects, in line with recent reports [27].

## 5. Conclusions

We reported an unprecedented mode of regulation of *TRIB1* downstream the oncogene *cMYC*, and we took advantage of genetically engineered PCa mouse models to provide experimental support for the role of the pseudokinase in the pathogenesis of this disease. 

## Figures and Tables

**Figure 1 cancers-12-02593-f001:**
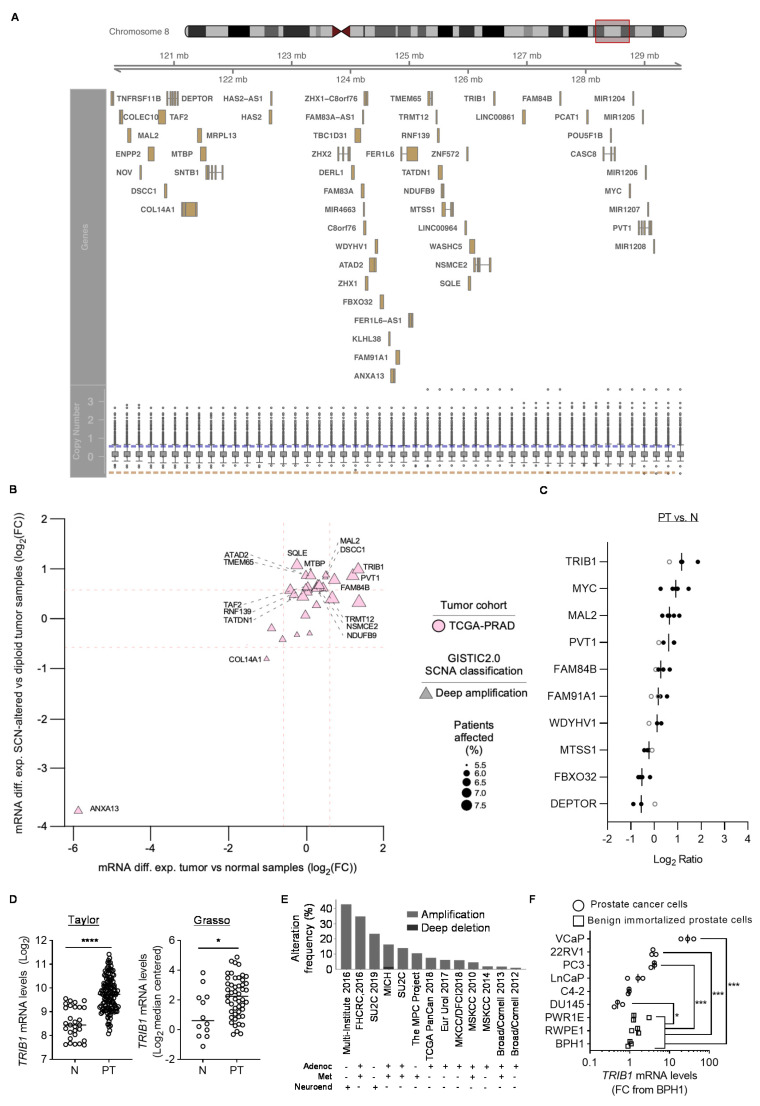
*TRIB1* is frequently amplified and overexpressed in PCa. (**A**) Copy number variation at *cMYC* locus. Overview of the genes located in *cMYC* amplicon in prostate cancer. Boxplots represent the distribution of copy number variation per gene in the TCGA datasets (492 specimens), given as GISTIC2 scores (log2 (copy-number/2)). Blue and orange lines represent the thresholds for copy-gain and loss, respectively. (**B**) Genes contained in *cMYC* amplicon were defined by two FC values: (1) tumor vs. normal samples (*x*-axis) and (2) SCN-altered vs. diploid tumor samples (*y*-axis). Only those with an FDR fold change (FC)-associated value < 0.05 were plotted. Deep amplifications are represented with a triangle, and its size is proportional to the % of TCGA-PRAD patients carrying a deep *cMYC* locus amplification, as defined by the GISTIC2.0 algorithm. Gene symbols point to those with a significant differential expression between SCN-altered and diploid tumor samples (|Log2(FC)| > 0.58 and FDR-value < 0.05). (**C**) Waterfall plot depicting the expression of indicated genes in up to five prostate cancer datasets [40,46,47,48,49,50]. Each dot represents the differential mRNA abundance in primary tumors (PT) vs. non-cancerous prostate tissue (N) for a given dataset. Black dots indicate a significant difference in expression, whereas grey dots depict gene expression differences that are non-significant according to two-tailed Student’s *t*-test. (**D**) Gene expression analysis of *TRIB1* in two human prostate cancer datasets in normal (N) versus primary tumors (PT). Data were extracted from Cancertool. Each dot indicates one individual. *, *p* < 0.05; ****, *p* < 0.0001. Statistics: two-tailed Mann–Whitney U test. (**E**) Copy number alteration analysis of the indicated prostate cancer studies. Data were extracted from cBioPortal. Adenoc: adenocarcinoma (localized); Neuroend, neuroendocrine tumor; Met, metastasis. (**F**) Relative *TRIB1* mRNA expression measured by RT-qPCR in benign immortalized prostate (BPH1, RWPE1 and PWRE1) versus prostate cancer (DU145, PC3, C4-2, 22RV1 and VCap) cell lines. Each dot indicates one biological replicate. *, *p* < 0.05; ***, *p* < 0.001. All values are normalized to BPH1. *GAPDH* was employed for normalization. FC: fold change. Statistics: two-tailed Student’s *t*-test.

**Figure 2 cancers-12-02593-f002:**
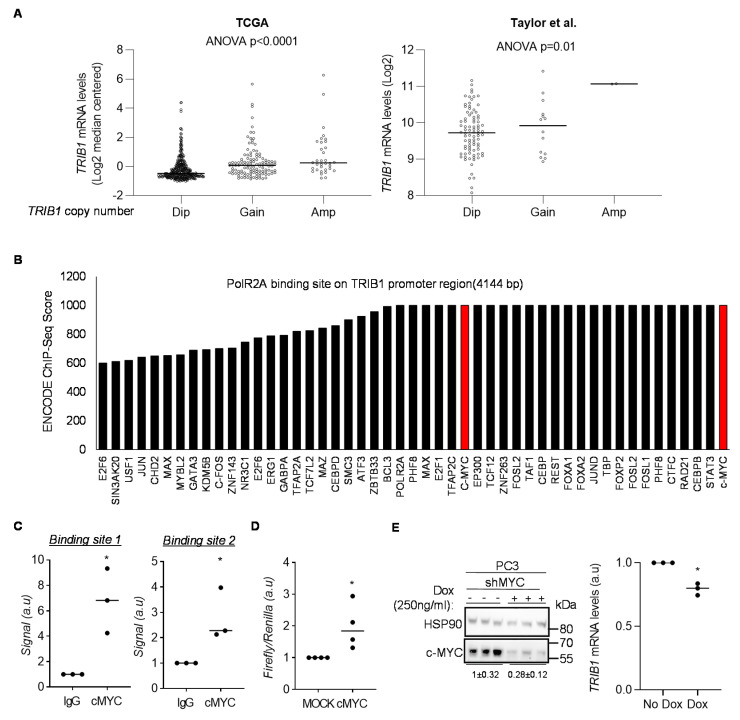
cMYC is a transcriptional regulator of *TRIB1*. (**A**) Association of *TRIB1* copy number to mRNA expression in TCGA and Taylor datasets. Data were extracted from cBioPortal. Dip, diploid; Amp, amplified. (**B**) Transcription factors with ENCODE binding score higher than 600 present in PolR2A binding region (4144 bp) on *TRIB1* regulatory region. Red bars indicate two cMYC binding sites with ENCODE binding score 1000/1000 in this region. Data were extracted from https://genome.ucsc.edu. (**C**) ChIP-RT-qPCR analysis of cMYC binding to *TRIB1* promoter region. Two cMYC binding sites on *TRIB1* regulatory region were selected based on the ENCODE3 binding score (1000/1000) and subject to ChIP analysis in PC3 cells. Quantitation of amplified immunoprecipitated DNA is indicated relative to input IgG. Each dot represents one biological replicate; a.u. = arbitrary unit. Statistics: one-tailed one-sample *t*-test. *, *p* < 0.05. (**D**) Dual-luciferase reporter assay, using *TRIB1* promoter and ectopic cMYC expression. Each dot represents one biological replicate. MOCK: empty vector. Statistics: one-sample *t*-test. *, *p* < 0.05. (**E**) Impact of inducible *cMYC* silencing on *TRIB1* mRNA expression in PC3 cells. Left panel shows cMYC downregulation upon activation of the shRNA with 250 ng/mL of doxycycline for six days (densitometry of cMYC relative to HSP90 is indicated, mean ± standard error), and right panels depict *TRIB1* mRNA abundance (values are normalized to no dox); a.u. = arbitrary unit. Statistics: one-sample *t*-test. *, *p* < 0.05. Uncropped western blot figure in Appendix A.

**Figure 3 cancers-12-02593-f003:**
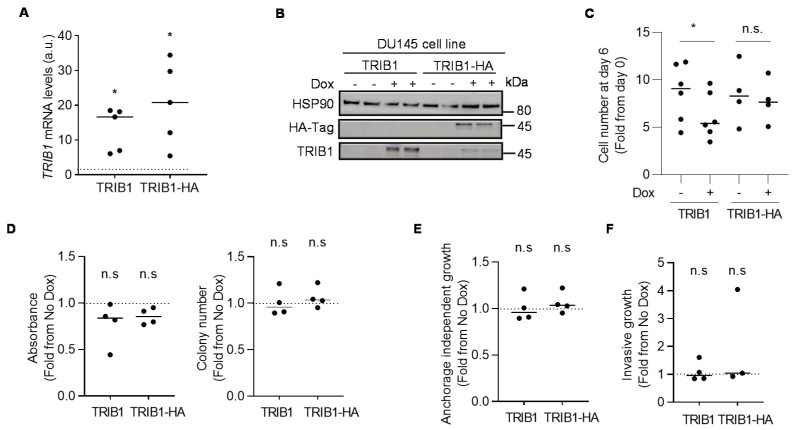
Ectopic expression of TRIB1 in DU145 cells does not influence tumor cell function. *TRIB1* mRNA (**A**) and protein expression (**B**) were measured by using RT-qPCR and Western blot, respectively. Each dot represents one biological replicate in the RT-qPCR data. HSP90 serves as a housekeeping control for Western blot analysis. TRIB1-HA: TRIB1 protein with C-terminal HA-tag. *β-ACTIN* was used for normalization in RT-qPCR. Dashed line shows normalization of values to non-induced samples in RT-qPCR; a.u. = arbitrary unit. Statistics: one-sample *t*-test. *, *p* < 0.05. Uncropped western blot figure in Appendix A. (**C**) DU145 cell growth was measured by crystal violet staining at day zero, and after three or six days post-doxycycline induction. Each dot represents one biological replicate; n.s. = statistically not significant; a.u. = arbitrary unit. Statistics: paired Student’s *t*-test. *, *p* < 0.05. (**D**) Evaluation of the effect of TRIB1 overexpression on the clonal growth. Colonies formed by DU145 cells were counted, and the crystal violet absorbance was measured after 14 days (left and central panels). Dashed line shows normalization of values to non-induced samples. Each dot represents one biological replicate; n.s. = statistically not significant; a.u. = arbitrary unit. Statistics: one-sample *t*-test. (**E**) Analysis of the anchorage independent growth of DU145 cells upon overexpression of TRIB1. Colonies were counted after three weeks of seeding. Each dot represents one biological replicate. Dashed line shows normalization of values to non-induced samples; n.s. = statistically not significant; a.u. = arbitrary unit. Statistics: one-sample *t*-test. (**F**) Analysis of the 3D invasive growth of DU145 cells upon overexpression of TRIB1. Each dot represents one biological replicate. Dashed line shows normalization of values to non-induced samples; n.s. = statistically not significant; a.u. arbitrary unit. Statistics: one-sample *t*-test.

**Figure 4 cancers-12-02593-f004:**
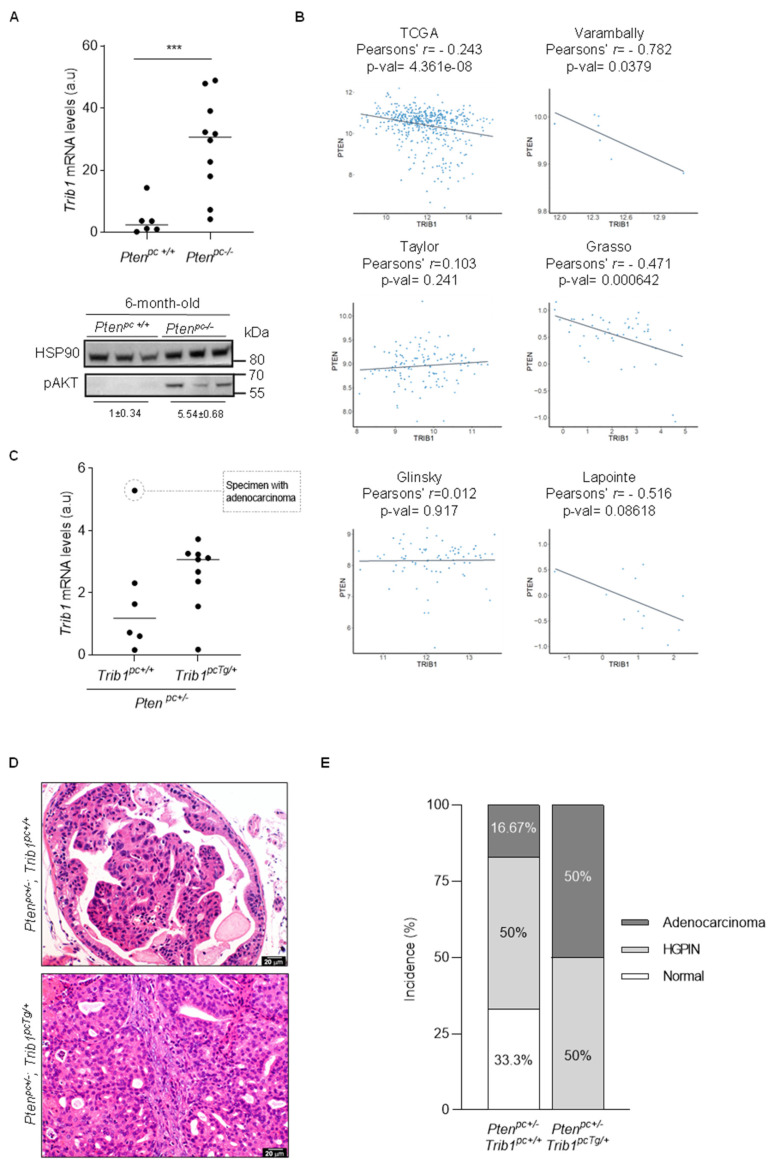
Transgenic *Trib1* expression promotes prostate cancer pathogenesis. (**A**) Measurement of the relative gene expression level of *Trib1* by RT-qPCR in anterior prostate (AP) lobe extracted from six-month-old *Pten^pc+/+^* (*n* = 6) and *Pten^pc−/−^* (*n* = 10) mice. Values are normalized to *Gapdh*; a.u. = arbitrary unit. Lower panel illustrates the increased in AKT serine 473 phosphorylation as a control of *Pten* deletion (densitometry of pAKT relative to HSP90 is indicated, mean ± standard error). Statistics: two-tailed Mann–Whitney U test. ***, *p* < 0.001. Uncropped western blot figure in Appendix A. (**B**) Correlation analysis and linear regression lines of *TRIB1* with *PTEN* mRNA levels in primary prostate cancer patient datasets. The corresponding Pearson’s r and *p*-values of the analysis are shown. (**C**) Evaluation of *Trib1* mRNA level by RT-qPCR in 15–17-month-old Pten^pc+/−^/Trib1^pc+/+^ (*n* = 6) and Pten^pc+/−^/Trib1^pcTg/+^ mice (*n* = 9). Each dot is representative of one individual mouse. Values are normalized to Gapdh; a.u. = arbitrary unit. (**D**) H&E staining of AP tissue from 15–17-month-old mice representative of high-grade prostatic intraepithelial neoplasia (HGPIN) in Pten^pc+/−^/Trib1^pc+/+^ and adenocarcinoma in Pten^pc+/−^/Trib1^pcTg/+^ mice. (**E**) Pathological analysis of prostate tissue isolated from 15–17-month-old Pten^pc+/−^/Trib1^pc+/+^ (*n* = 6) and Pten^pc+/−^/Trib1^pcTg/+^ (*n* = 8). The data correspond to the prostate lobe with most significant phenotype. Phenotypes: high-grade prostatic intraepithelial neoplasia (HGPIN) and prostate adenocarcinoma.

**Figure 5 cancers-12-02593-f005:**
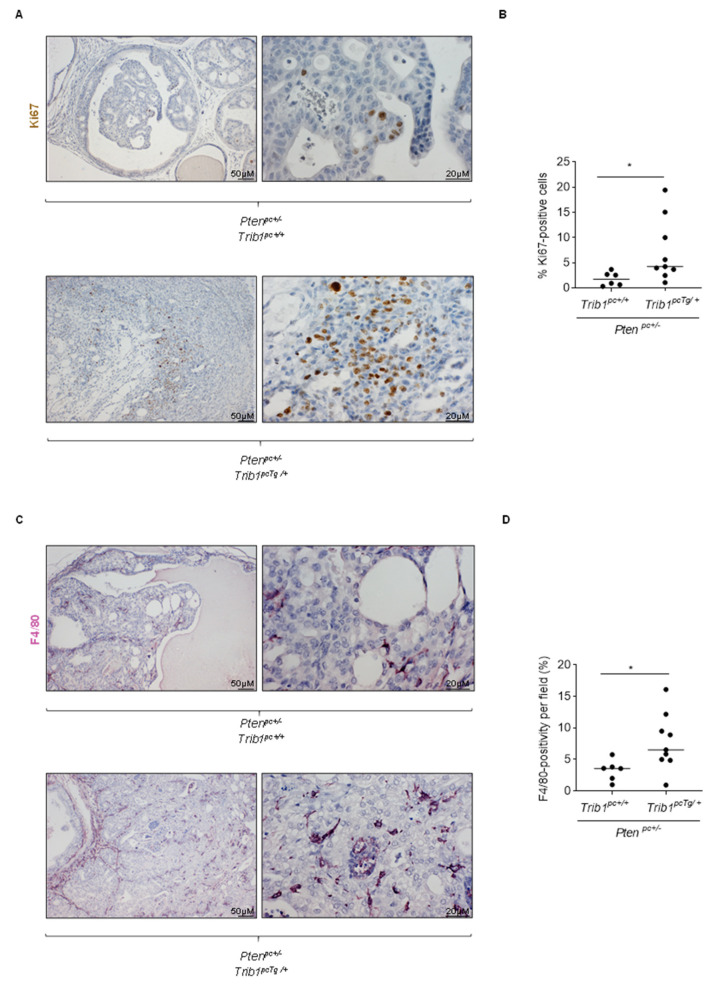
Transgenic *Trib1* overexpression in the prostate epithelium promotes cell proliferation and macrophage infiltration. (**A**) Analysis of tumor cell proliferation by immunostaining of the proliferation marker Ki67 in prostate tissue sections isolated from Pten^pc+/−^/Trib1^pc+/+^ (*n* = 6) and Pten^pc+/−^/Trib1^pcTg/+^ mice (*n* = 9). Representative images are presented at two different magnifications. (**B**) The percentages of Ki67 positive cells were quantified by using Image J, relative to the total number of cells. Data represent five 20X-field per tissue. Statistical test: two-tailed Mann–Whitney U test. *, *p* < 0.05. (**C**) Macrophage infiltration was assessed by immunostaining analysis of mouse macrophage marker F4/80 in prostate tissue sections isolated from Pten^pc+/−^/Trib1^pc+/+^ (*n* = 6) and adenocarcinoma in Pten^pc+/−^/Trib1^pcTg/+^ (*n* = 9) mice. Representative images are presented at two different magnifications. (**D**) The number of F4/80 positive cells per 20X field was quantified by using Image J. Data represent the average of five 20X-field per mouse. Statistical test: two-tailed Mann–Whitney U test. *, *p* < 0.05.

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
