# Peer review of "Genomic and Functional Regulation of TRIB1 Contributes to Prostate Cancer Pathogenesis"

_cancers, 2020, doi:10.3390/cancers12092593_

Round 1

Reviewer 1 Report

I reviewed the previous version of this manuscript submitted to “Cancers” and I was generally positive about it, with several improvements suggested to the authors. My suggestions were addressed as those for “Reviewer 4” in the rebuttal letter. They were addressed fine, taking to account methods/model/availability limitations. I still have a doubt as to why e.g. the authors did not show another human prostate cell line to confirm results in Fig.2D, as suggested, and used TNBC MDA-MB-231 cell line instead. It looks like a situation of a sort of “did not work as expected in prostate cancer, let’s take something that works, e.g. breast cancer”. On the other hand – as I wrote in my previous review – my propositions were suggestions, not strict requests, and I’m aware of problems/limits some of the suggested models may produce. Given the overall improvements to the manuscript, replying to – sometimes I think even too radical – requests of other reviewers of the first manuscript version, in my opinion the manuscript is OK for publication in the current state.

Author Response

I reviewed the previous version of this manuscript submitted to “Cancers” and I was generally positive about it, with several improvements suggested to the authors. My suggestions were addressed as those for “Reviewer 4” in the rebuttal letter. They were addressed fine, taking to account methods/model/availability limitations. I still have a doubt as to why e.g. the authors did not show another human prostate cell line to confirm results in Fig.2D, as suggested, and used TNBC MDA-MB-231 cell line instead. It looks like a situation of a sort of “did not work as expected in prostate cancer, let’s take something that works, e.g. breast cancer”. On the other hand – as I wrote in my previous review – my propositions were suggestions, not strict requests, and I’m aware of problems/limits some of the suggested models may produce. Given the overall improvements to the manuscript, replying to – sometimes I think even too radical – requests of other reviewers of the first manuscript version, in my opinion the manuscript is OK for publication in the current state.

We would like to thank the reviewer for his/her help in improving and refining this study.

As per his/her comment on our validation of MYC-TRIB1 regulation in TNBC, we chose to use this model system for two reasons:

  1. We perceived that the extension of our observations to another cancer type could be of interest for a broad readership.
  2. In the timeframe in which the experiments were performed, we could not obtain a significant downregulation of cMYC using shRNA in DU145 cells.

Reviewer 2 Report

no further comments

Author Response

no further comments

We would like to thank the reviewer for his/her help in improving and refining this study.

Reviewer 3 Report

Major concerns:

  1. In Fig 3C, the authors stated in the text (line 378) that “Inducible expression of TRIB1 in DU145 cells did not alter consistently cell proliferation”. However, in Fig 3C, adding the Dox to induce TRIB1 in DU145 cells did decrease cell numbers as compared to the control which is without Dox. The data also reached a statistical different as the author indicated in the figure with an asterisk. It does not make any sense that the authors concluded without really based on their presented data here.
  2. In the point-to-point response from the authors, they claimed that one sample t-test is more restrict. However, in the some figures, especially for these from the cell line experiments such as Fig 3, they used the control (without Dox) and the experiment (with Dox to induce TRIB1) sets, which means that they always have the control group to compare with their experimental group. Therefore, the one sample t-test method is inappropriate.
  3. The authors performed the xenograft mouse experiments in the revised manuscript but trying to claim the discrepancy between the transgenic mouse models and cell lines is due to the stroma. However, the xenograft mouse experiments still did not support their argument here because the interaction of the human implanted prostate cancer cells with the mouse stromal cells is restrained and doesn’t mimic the real tumor-stroma interaction in the transgenic mouse models which have mouse tumors interacted with mouse stroma. Please explain why this new piece of data helps with strengthening the original claim.
  4. Although the authors provided the WB and claimed that there is no available antibody to recognize the endogenous TRIB1 for their transgenic mice and also for the human tissue samples, this is very critical point to argue why the inconsistent result between the in vitro and in vivo experiments. How do you prove that your transgenic mice had the effect are due to the overexpressed TRIB1 if the protein level is not detected and at the same time, the mRNA level of RIB1 is just merely over its basal level? If the results between the in vitro and in vivo experiments were consistent, there will not be any suspension at all. However, this is not the case in this manuscript, therefore, it is way critical to provide this evidence for this manuscript.

Minor issues/concerns:

  1. In the Statistics and Reproducibility subsection under the Methods section (line 232), the authors wrote “For each independent in vitro experiment, at least three technical replicates and a minimal number of three experiments were performed to ensure adequate statistical power”. Please indicate the exact number of biological replicates for obtaining the result in the figure legend for each figure.
  2. The supplementary table 1 content is confusing. For example, in the column of Forward 5’-3’, the authors listed the sequence. Then the column should be named forward probe sequence. Same issue with the probe column in this table. In addition, please fill in the information for the probes for GAPDH.
  3. In supplementary table 5, the authors included the GeneSymbol column but left it blank. Is this a mistake or on purpose? If it’s a mistake, please put all genes in the table. If it is on purpose, please explain the specific purpose.

Author Response

We would like to thank the reviewer for his/her help in improving and refining this study.

Major concerns:

  1. In Fig 3C, the authors stated in the text (line 378) that “Inducible expression of TRIB1 in DU145 cells did not alter consistently cell proliferation”. However, in Fig 3C, adding the Dox to induce TRIB1 in DU145 cells did decrease cell numbers as compared to the control which is without Dox. The data also reached a statistical different as the author indicated in the figure with an asterisk. It does not make any sense that the authors concluded without really based on their presented data here.

The statement of our conclusion is that the results obtained where not consistent using different TRIB1-overexpressing constructs. This fact added to the complementary cellular assays in which ectopic TRIB1 expression did not alter cellular responses suggested that the effect of this gene on the assays monitored is negligible. In the revised version we have amended our statement, including further detail:

“Inducible TRIB1 expression (C-terminal HA-tagged or untagged) in DU145 cells did not alter consistently cell proliferation (Fig. 3C). The expression of untagged TRIB1 significantly reduced cell number, whereas the C-terminal tag form of the pseudokinase did not exert any effect. Neither of the constructs altered colony formation (Fig. 3D), anchorage-independent growth (Fig. 3E) or invasive growth in three-dimensional systems (Fig. 3F).”

  1. In the point-to-point response from the authors, they claimed that one sample t-test is more restrict. However, in the some figures, especially for these from the cell line experiments such as Fig 3, they used the control (without Dox) and the experiment (with Dox to induce TRIB1) sets, which means that they always have the control group to compare with their experimental group. Therefore, the one sample t-test method is inappropriate.

One sample T-test is meant for experimental settings in which one of the groups has no variance (all the values are the same). This happens in assays in which the treatment values are relative to those of the control. We would like to insist on the fact that the one sample T test is not only more adequate, it is also more restrictive than the regular Student T-test.

  1. The authors performed the xenograft mouse experiments in the revised manuscript but trying to claim the discrepancy between the transgenic mouse models and cell lines is due to the stroma. However, the xenograft mouse experiments still did not support their argument here because the interaction of the human implanted prostate cancer cells with the mouse stromal cells is restrained and doesn’t mimic the real tumor-stroma interaction in the transgenic mouse models which have mouse tumors interacted with mouse stroma. Please explain why this new piece of data helps with strengthening the original claim.

The xenograft experiment was performed and included to provide a more comprehensive perspective of the activity of TRIB1 in different model systems. We emphasize the consistency of our results with the recent study of Liu et al., Cell Signal 2019, thus supporting the notion that our strategy and results in vitro and in vivo are not the consequence of experimental artifacts.

  1. Although the authors provided the WB and claimed that there is no available antibody to recognize the endogenous TRIB1 for their transgenic mice and also for the human tissue samples, this is very critical point to argue why the inconsistent result between the in vitro and in vivo experiments. How do you prove that your transgenic mice had the effect are due to the overexpressed TRIB1 if the protein level is not detected and at the same time, the mRNA level of RIB1 is just merely over its basal level? If the results between the in vitro and in vivo experiments were consistent, there will not be any suspension at all. However, this is not the case in this manuscript, therefore, it is way critical to provide this evidence for this manuscript.

We consider that technical limitations of antibody availability prevent us from answering this point. The positive response from the other 3 reviewers is a measure of the technical quality of the manuscript to overcome this aspect.

Minor issues/concerns:

  1. In the Statistics and Reproducibility subsection under the Methods section (line 232), the authors wrote “For each independent in vitro experiment, at least three technical replicates and a minimal number of three experiments were performed to ensure adequate statistical power”. Please indicate the exact number of biological replicates for obtaining the result in the figure legend for each figure.

The figure legend indicates in all cases biological replicates, since technical replicates have no statistical value. We have included a new statement in the statistics section of the methods:

“For each independent in vitro experiment, at least three technical replicates were used and a minimum number of three experiments were performed to ensure adequate statistical power (the number of biological replicates is indicated in the figure legends).”

  1. The supplementary table 1 content is confusing. For example, in the column of Forward 5’-3’, the authors listed the sequence. Then the column should be named forward probe sequence. Same issue with the probe column in this table. In addition, please fill in the information for the probes for GAPDH.

The reviewer is right. The column heading has been corrected.

  1. In supplementary table 5, the authors included the GeneSymbol column but left it blank. Is this a mistake or on purpose? If it’s a mistake, please put all genes in the table. If it is on purpose, please explain the specific purpose.

The reviewer is right. This table has been corrected.

Reviewer 4 Report

The authors adressed all comments appropriately.

Author Response

The authors adressed all comments appropriately.

We would like to thank the reviewer for his/her help in improving and refining this study.

Round 2

Reviewer 3 Report

Although the authors provided the point-to-point response to my questions especially in the major issue area, none of their answers critically addressed my concerns. This may due to their unwillingness to improve the quality of the manuscript at this current stage. The major concerns remain unaltered in my viewpoint since there are no scientific basis from the authors' side to really seek the answers for these major concerns.

In addition, the authors insist to use the one sample test for their in vitro data which apparently have 2 groups namely control and experimental groups. The reason that the authors insisted to use one sample test because they set their control value from different biological repeats always as 1 no matter the original readings for the control. This is what most labs do not do because even the control group should still have the SE among different biological repeats. This is the major reason that you don't see publications in this field use the one sample test at all which can result in misintepretation of the obtained data. 

This manuscript is a resubmission of an earlier submission. The following is a list of the peer review reports and author responses from that submission.

Round 1

Reviewer 1 Report

In the study of Shahrouzi et al, the authors describe TRIB1 to play a functional role in prostate cancer and to be amplified in a subset of 8q24 amplified prostate cancers. In silico data and in vivo models reveal associations between TRIB1, cMYC and PTEN which is of interest as cMYC and PTEN belong to frequently altered genes in prostate cancer. In general, results of this study provide important data about TRIB1 which has to further characterised. 

There are some points the authors should consider during revisions of this manuscript:

  • In the introduction, the authors should give more detailed information of genetic cMYC alterations in prostate cancer (frequency of amplification; which kind of tissue: primary PCa vs. metastasis vs. CRPC)
  • Similar in the results part, the authors should be more precise by giving information about the tissue types (primary PCa vs metastatic tumors). Instead of citing the authors’ name of the studies provided by cBioportal (Figure 1C, 1D), it would me more informative to see which tissue type is presented in each plot
  • cMYC silencing reduces TRIB1 levels for approximately 20%. Might this be explained by the concomitant TRIB1-up-regulation by amplification in PC3 cells? cMYC knockdown should also be performed in cells harbouring TRIB1 wild-type status to investigate this (in)dependence.
  • The differentiation between focal and invasive adenocarcinoma should be avoided or is misnamed at this point. Adenocarcinoma is invasive per definition and can only be differentiated from non-invasive epithelial changes including HGPIN.
  • Does TRIB1 modulation have any effect on the growth of invasive adenocarcinoma?
  • In this study, PTEN and TRIB1 are described to have oncogenic effects during tumor initiation. Of note, both PTEN and cMYC alterations are mainly seen in advanced, metastatic and castration-resistant PCa in humans. This seems controversial and should be explained by the authors. 
  • In addition to Ki67 staining, an alternative method should be added to measure the effects of TRIB1 on proliferation/tumor growth. 
  • It would be interesting to get information about the expression of TRIB1 in human prostate cancer tissues, also compared to benign prostate epithelium and HGPIN.
  • The authors should consider that the differentiation between low-grade and high-grade PIN is not used for human PIN. While HGPIN has to reported, in particular in prostate needle biopsies, LGPIN should not be diagnosed. The authors might exclude this differentiation or discuss this important point as there is a lack of translational relevance. 

Reviewer 2 Report

Shahrouzi et al have shown the increased gene expression of TRIB1 in the existing reported prostate cancer patient databases including the TCGA, the commonly used gene database in the cancer research field as well as in the commonly used human prostate cancer cell lines. They also showed that manipulation of TRIB1 including overexpression or knockdown in culture human prostate cancer cell lines did not affect cell growth, colony forming ability, anchorage-independent growth and cell invasion. However, when expression of TRIB1 specifically in prostate epithelial cells, at the same time, combined with loss of PTEN, it accelerated these PTEN-/- cells to become invasive prostate cancer cells. The authors were correlated this promoted effect of TRIB1 in their mouse model to elevated cell proliferation (increased Ki-67) and macrophage infiltration (increased F4/80). However, the relationship between macrophage infiltration and TRIB-1 caused prostate cancer formation in the PTEN-/- background was not explicitly address or explained at all.    

Although the authors presented many figures and graphs including supplemental figures and tables, they did not do a good job especially in the Results section to detail their experimental results. Instead, they used a very general way to go through their each figure especially when a particular portion of the result showed the opposite conclusion as what they want to claim. For example, in lines of 230-231, the author wrote” and its consequence on gene expression in PCa cell lines was confirmed by real time quantitative PCR (RT-qPCR, Fig. 1E). In Fig. 1E, the TRIB1 mRNA levels in prostate cancer cell lines C4-2 and LnCaP were actually no difference as compared to the non-tumorous prostate epithelial cell lines including BPH1, RWPE1 and PWR1E. This is exactly not to confirm what the authors wrote in their Results section. There are many similar serious mistakes through the whole article which makes this manuscript suspicious regarding what the authors were trying to claim through their texts. Furthermore, the data shown is not well organized especially regarding the detailed information in the corresponding figure legends and some figures are not in a consistent format especially in mRNA levels throughout the entire manuscript.

Another very serious concern is that if the authors claimed that manipulation of TRIB1 didn’t have any effect on cancer cell growth, colony forming ability, anchorage-independent growth nor cancer cell invasion in their in vitro system (Fig 3 and their supplemental fig for PC3 cells), how come their TRIB1 transgenic mouse model showed the effect on cell proliferation as shown in Ki-67 and also promoted cancer invasion (Fig 4-5). Both DU-145 and PC3 cells are loss of PTEN (DU-145: PTEN+/-; PC3: PTEN-/-). Therefore, the most possible explanation for the results in their mouse model is due to the artificial effects of overexpression of TRIB1.

Other major issues/concerns include:

  1. In Fig 1D, the TRIB1 alteration rate is in a wide range from almost down to 0 to over 40% in different prostate cancer databases. The authors didn’t intepretate what it means nor mention this wide range in their Results section at all even though they choose to show this piece of data.
  2. For many figures, the statistics was done using One-Sample t-test. It needs to be evaluated using the student t-test. Also the authors wrote the SEM was indicated but this is not shown in any figures at all. Only the mean is shown in each figure.  
  3. In Fig. 3, cell numbers, colony numbers and invaded cell numbers in these charts were apparently way too low. This can counts for significant experimental errors and give false results.
  4. In Fig 4C, the mRNA levels from the control transgenic to TRIB1 transgenic mouse prostate only increased from 1 fold to less than 3 fold. Since a small increase in mRNA doesn’t guarantee an increase in protein expression, it will be very important to know the protein levels of TRIB1 in control transgenic vs in TRIB1 transgenic mouse prostate tissues. This can be shown by WB or IHC.
  5. In Fig 5, it is almost impossible to see the Ki-67 and F4/80 staining in the presented images. It is necessary to provide the zoom-in fields.
  6. Although the authors originally identified TRIB1 using the human prostate cancer databases, still it is important to show the expression of TRIB1 proteins in human prostate cancer tissues v.s. normal prostate (adjacent normal) tissue for validating their results.

Reviewer 3 Report

Grammar needs some serious attention. E.g. Line 52 “Albeit the implementation of innovative therapies for recurrent PCa, emergence of metastasis in these patients is frequent and represents a major risk of mortality by this disease.”

Line 435 “Interestingly, other oncogenic insults relevant to PCa also affect TRIB1 gene expression.”

I don’t understand this.

There are many more issues throughout the manuscript.

Laeukemogenesis – line 70. Should be leukemogenesis.

The evidence of PTEN negative correlation with TRIB1 in Figure 4B is weak. Very few samples in Varambally and Lapointe. R <0.25 suggests no linear association. What microarray probes were used for this data; some genes have multiple probes that will show different results

Analysis of Trib1 mRNA, figure 4C. only 4 mice were used as ‘controls’ vs 8 mice for Tg/+ ? Why? More mice should be used in this control set.

Figure 4D – representative data shown but analysis was performed on more. These images should be in supplementary dataset and annotated.

Supplementary figures – blot images are over-processed showing saturated signal or lack of background.

The method of cutting blots to probe for a specific molecular weight is disappointing and additional bands are unexplained. E.g. S4A, supp fig 5.

NO PTEN expression performed for Supp Figure 4 c-MYC expression blot. How to know if these were down regulated?

qPCR performed with normalisation to GAPDH… this is inadequate. GAPDH may be altered in prostate cancer and use of a single gene is unreliable. The use of multiple endogenous controls is recommended, e.g. HPRT1, ACTB, GAPDH

Reviewer 4 Report

The study by Shahrouzi et al. provides an important insight into the role of TRIB1 as an oncogene in prostate cancer, its cooperation with CMYC and a PC mouse model with overexpressed TRIB1. As such, the study is worth being published in Cancers and is a valuable source of information extending the knowledge on the oncogenic role of TRIB1. Obviously the study would benefit from more research on an exact mechanism of the TRIB1 action in PC in the cell non-autonomous context, but I assume this was out of the scope of the study, and hopefully will be addressed in the future papers of the team. However there are several components that could be added to the scope of the current manuscript, which may extend and strengthen it. These are in my opinion not required for publication, but well worth considering to be included by the authors in a minor revision of the manuscript:

  1. The most important issue which could be addressed is the discrepancy between human and mouse context vs. the cell-autonomous and non-autonomous context. The experiments in the human context were done in cell lines in vitro where there was no effect of TRIB1 silencing/overexpression while the experiments in the in vivo mouse model mouse model showed clear oncogenic effects of TRIB1. But is this for sure only due to the cell-autonomous and non-autonomous context, and not due to the human vs mouse neoplasia differences? This could be addressed by – on one hand – performing TRIB1 overexpression/silencing experiments in a mouse PC model cell line/fibroblasts in vitro (to see whether they match the results in human cell lines) and – on the other hand – performing TRIB1 staining in human patient PC tissues (to confirm the increase of TRIB1 as well as correlate it with grade/aggressiveness and MYC status ) and Kaplan-Meier analysis in PC patients to correlate survival with TRIB1/MYC expression levels. Perhaps at least some of these experiments' results could be added to the revised manuscript.
  2. Is it possible to extend the number of human cell lines tested in Figure 3 and Supplementary Figure 3 with one more high and one more low TRIB1 level cell line? This would strengthen the results in the human system beyond just single cell lines.
  3. Is it possible to perform a test of dependence of the TRIB1 expression on the CMYC overexpression (Fig. 2D) in at least one human PC cell line – such as DU145 – with a low level of TRIB1? This would additionally confirm this induction beyond the HEK293 cell line, which is not PC.